# Precarious Employment and Psychosocial Hazards: A Cross-Sectional Study in Stockholm County

**DOI:** 10.3390/ijerph182111218

**Published:** 2021-10-26

**Authors:** Signild Kvart, Johanna Jonsson, Theo Bodin, Carin Håkansta, Bertina Kreshpaj, Cecilia Orellana, Per-Olof Östergren, Lotta Nylén, Nuria Matilla-Santander

**Affiliations:** 1Unit of Occupational Medicine, Institute of Environmental Medicine, Karolinska Institutet, 11365 Stockholm, Sweden; johanna.jonsson@ki.se (J.J.); theo.bodin@ki.se (T.B.); carin.hakansta@ki.se (C.H.); bertina.kreshpaj@ki.se (B.K.); cecilia.orellana@ki.se (C.O.); nuria.matilla.santander@ki.se (N.M.-S.); 2Center for Occupational and Environmental Medicine, Stockholm Region, 11365 Stockholm, Sweden; 3Working Life Science, Karlstad Business School, Karlstad University, 65188 Karlstad, Sweden; 4Social Medicine and Global Health, Department of Clinical Sciences Malmö, Lund University, 20502 Malmö, Sweden; per-olof.ostergren@med.lu.se; 5Department of Global Public Health, Karolinska Institutet, 17177 Stockholm, Sweden; lotta.nylen@ki.se; 6Academic Primary Health Care Center, Stockholm Region, 11365 Stockholm, Sweden

**Keywords:** precarious employment, psychosocial work environment, work environment hazards, employment conditions

## Abstract

Precarious employment (PE) has been linked to adverse health effects, possibly mediated through psychosocial hazards. The aim of this cross-sectional study is to explore if higher levels of PE are associated with psychosocial hazards (experiences of violence, sexual harassment, bullying, discrimination, high demands, and low control) and to explore gender differences in these patterns. The study is based on survey- and register data from a sample of 401 non-standard employees in Stockholm County (2016–2017). The level of PE (low/high) was assessed with the Swedish version of the employment precariousness scale (EPRES-Se) and analysed in relation to psychosocial hazards by means of generalized linear models, with the Poisson family and robust variances. After controlling for potential confounders (gender, age, country of birth, and education), the prevalence of suffering bullying (PR 1.07, 95% CI: 1.01–1.13) and discrimination (PR 1.52, 95% CI: 1.00–2.32) was higher among individuals with a high level of PE. Regarding the demand/control variables, a high level of PE was also associated with low control (PR 1.59, 95% CI: 1.30–1.96) and passive work (the combination of low demands and low control) (PR 1.60, 95% CI: 1.23–2.08). Our findings suggest that workers in PE are more likely to experience psychosocial hazards, and these experiences are more prevalent among women compared to men. Future longitudinal studies should look further into these associations and their implications for health and health inequalities.

## 1. Introduction

Across developed western countries, labour markets have undergone several changes during the last 50 years. The globalisation of the economy together with economic crises, technological innovations, and neoliberal economic policies have led to a ‘flexibilisation’ of the labour market [1]. A central part of this development is the increasing replacement of the standard employment relationship (SER) by ‘non-standard’ employment arrangements [2] such as involuntary part-time, on-demand work, and ‘gig’-arrangements, including crowd work and other digital platform work (ILO, 2016).

Some, but not all, non-standard employment relationships are characterized by unfavourable conditions for the worker, such as low or unpredictable earnings and a general shift of risk from employer to employee [3,4,5]. Among non-standard contracts in Sweden, there has been a gradual increase in the most unfavourable type: the ‘on-demand’ contract, which means employment by the hour and high unpredictability for the employee. Simultaneously, there has been a decrease in other types of non-standard contracts where the conditions are more similar to standard employment, such as substitute contracts [6,7]. One can therefore argue that non-standard contracts have become more precarious in Sweden during the last decades [7]. Research that focuses on unfavourable conditions sometimes use the term precarious employment (PE) instead of non-standard employment. PE is a multidimensional construct which encompasses various dimensions of disadvantage, such as employment insecurity, income inadequacy, and lack of workplace rights and protection (e.g., protection against unjustifiable dismissal, discrimination, or harassment) [8]. Most research conducted on PE is based on unidimensional measurements of PE, such as contract type or duration. PE has been linked to adverse health outcomes such as occupational injuries and poor mental health [9,10]. Psychosocial work environment hazards have been linked with several health outcomes [11], and it has been suggested that this may be one of the pathways linking PE to adverse health [12,13], but empirical evidence is lacking.

Psychosocial hazards such as job strain from the combination of high demands and low control have received some attention in studies on agency work, temporary, and on-call employment, but the evidence has been inconclusive [14,15]. Other psychosocial hazards such as violence/threats, bullying, harassment, and discrimination are less often studied, but in a study where multiple dimensions of PE were considered, precarious employees were found to be almost twice as likely to experience violence and harassment [16].

Moreover, considering the vertical and horizontal labour market segregation of men and women, it is important to keep a gender perspective when studying PE. Previous research has shown that women have a disadvantaged position in most dimensions of job quality [17], and that women and men are exposed to different types of work environment hazards [18,19]. It has also been argued that women and men may be differentially affected by these experiences [20,21].

The primary aim of the study is to explore if workers in high PE, compared to low PE, experience more violence/threats, bullying, sexual harassment, discrimination, and high demands/low control. A secondary aim is to explore gender differences in these patterns.

The research hypothesis is that employees with a higher level of PE are more likely to experience psychosocial hazards and, moreover, that these experiences differ for women and men with regard to the type of hazard.

The contribution of this study is a novel and more relevant definition of PE compared to previous studies using proxy indicators. Psychosocial work environment hazards in relation to PE as a multidimensional construct have rarely been studied, especially taking the gender perspective into account (see [13] for a recent exception). To the best of our knowledge, this is the first Swedish study of its kind.

## 2. Materials and Methods

### 2.1. Study Design and Data Collection

This cross-sectional study is based on survey data collected within the PREMIS (Precarious Employment in Stockholm) project (2016–2017). PREMIS is a research project aimed at studying occupational health risks in a sample of non-standard employees. The PREMIS research is based on survey data linked with register data.

Recruitment and data collection took place between November 2016 and May 2017, and the response rate to the online questionnaire was 62%. The study participants (*n* = 483) were recruited using web-based respondent-driven sampling (webRDS, a non-probability chain-referral sampling method). In brief, webRDS software was used for peer recruitment and data collection via an online survey. Recruitment was initiated by spreading information about the study through the reference group involved in the study and word of mouth; and through advertising in areas around Stockholm and online. Inclusion criteria for participants were: age 18–65 years, currently in non-standard employment (i.e., not in a permanent, full-time position), living and/or working within Stockholm County, and having a Swedish personal identification number. Exclusion criteria were: being voluntarily self-employed or voluntarily part-time employed, being a student or retired, or indicating an invalid personal identification number. After exclusion, 415 participants were eligible to be included. For this study, respondents with full valid information in the Swedish version of the employment precariousness scale (EPRES-Se) were included, leading to a final analytical sample of 401 individuals. Both the PREMIS study population and sampling procedure are described in a previous publication [22].

The survey included the Swedish version of the EPRES-Se, which is an instrument for measuring multiple dimensions of PE [23]. Apart from the EPRES-Se section, the questionnaire also contained items on health, work environment, sociodemographics, and current life situation. Some additional background data (level of education and country of birth) were obtained from Statistics Sweden, by linking the study participants personal identification number to the Longitudinal Integration Database for Health Insurance and Labour Market Studies (LISA, acronym in Swedish). The survey questionnaire and the EPRES-Se can both be found in the Supplementary Material of previous publications [22,23].

### 2.2. Study Variables

Exposure variable: level of PE (low/high) was operationalized using the EPRES-Se [23]. This instrument measures PE through 23 items grouped in 6 dimensions: (1) temporariness, e.g., contract type or duration; (2) wages, e.g., level of income and ability to sustain basic needs; (3) disempowerment, e.g., influence over working hours and salary; (4) vulnerability, e.g., fear of demanding better conditions and fear of being fired; (5) rights, e.g., sickness benefit and unemployment insurance; and (6) capacity to exercise rights, e.g., ability to take vacation days ‘without problem’.

Each item was first recoded into a 0–4 scale, and then the mean was calculated for each dimension. Thereafter, the overall EPRES-Se mean value was calculated for each study participant, a score that could theoretically range between 0 (not precarious) and 4 (most precarious). The median value for the overall population (EPRES-Se score: 1.92) was used as a cut-off to divide the study population in two groups, representing ‘low’ PE (EPRES-Se score range: 0.89–1.92) and ‘high’ PE (EPRES-Se score range: >1.92–3.07).

Outcome variables: The outcome variables were workplace violence, sexual harassment, bullying, discrimination, and job demands and control (with their combinations: high-strain job, low-strain job, active job, and passive job).

Variables related to workplace violence, sexual harassment, bullying, and discrimination were derived from the following item in the PREMIS survey: “Have you, during the last 12 months, been exposed to/suffered any of the following at work: Violence or threat of violence? Sexual harassment? Bullying? Discrimination due to gender? Discrimination due to ethnicity? Discrimination due to age?”. The response options were yes/no, and the respondents were asked to provide an answer to each of the sub questions. The three different types of discrimination were later collapsed into one (‘Discrimination’), in the main analysis, due to low prevalence.

First, to construct the four types of jobs based on the job demand/control model [24], variables related to demands and control were operationalized based on four questions of the PREMIS survey. The response options were dichotomized as: no (half the time, about 1/4 of the time, about 1/10 of the time) and yes (nearly all the time, about 3/4 of the time). The questions used to operationalize job demands were: “Is your work so stressful that you do not have time to talk or even think about something other than work?” and “Does work require your full attention and concentration?”. High job demands were determined by answering yes to both questions (otherwise classified as low job demands). The questions used to operationalize job control were: “Do you have the opportunity to determine your work pace?” and “Can you take short breaks at virtually any time?”. Low job control was determined by answering no to both questions (otherwise classified as high job control).

Second, the dichotomized variables (job demands and job control) were combined into the four types: high-strain job; combination of high demands and low control, low-strain job; combination of low demands and high control, active job; combination of high demands and high control, passive job; and combination of low demands and low control.

Covariates: the covariates considered in this study were obtained from the PREMIS survey (gender, age, occupation) and from the LISA register (country of birth and education). Gender was categorized as man or woman, age as 18–24, 25–29, 30–35, 36–62 years, country of birth as born in Sweden: yes or no, and education as three categories: high school, higher education less or equal to two years, and higher education more than or equal to three years. The categorisation of occupations as manual and non-manual was performed by translating the self-reported current occupation in the PREMIS survey into first digit level codes from the Swedish Standard Classification of Occupations for 2012 (a modification of ISCO-08), where codes 1–4 were classified as non-manual and codes 5–9 as manual [25]. The association between level of PE, psychosocial hazards, and other variables was explored in a directed acyclic graph (DAG), using Dagitty (a browser-based environment for creating, editing, and analyzing causal diagrams) [26]. Country of birth, gender, education, and age were identified as potential confounders (Figure 1), while occupation was identified as a covariate.

### 2.3. Statistical Analysis

First, frequencies and percentages with 95% confidence intervals (CI) for experiencing psychosocial hazards and covariates were calculated according to level of PE (low/high).

Second, the crude and adjusted prevalence ratios (PR and aPR) of experiencing psychosocial hazards according to levels of PE were explored by means of generalized linear models, with the Poisson family and robust variances, with the low PE group as reference [27]. The choice of Poisson models was based on the high prevalence of the outcomes studied (>10%) and thereby to avoid an overestimation of the effect. Adjustment was performed in two steps: first by adjusting for gender and age and thereafter including country of birth and education (fully adjusted model).

Third, gender-specific prevalence of experiencing psychosocial hazards was calculated according to level of PE. Due to the small sample size, PR and aPR stratified on gender were deemed infeasible.

Analyses were conducted using SPSS statistical software version 25 (IBM Corp, Armonk, NY, USA) and STATA 16.0 (Stata corporation, College Station, TX, USA).

## 3. Results

The sociodemographic characteristics of the study sample are presented in Table 1. In the high-PE group (compared to low PE), we found more of the youngest participants (68.1% among 18–24-year-olds), those with low education (56.4%), manual occupations (56.2%), and non-Swedish participants (60.5%).

Table 2 shows the prevalence and associations between level of PE and experiences of psychosocial hazards, with low PE as the reference group. The results show that experiences of psychosocial hazards were more prevalent in the high-PE group. This was true except for the prevalence of experiencing violence/threats (which was lower in high PE, 9.4% compared to 12.6%) and the prevalence of experiencing high demands (which was similar in both low and high PE).

After controlling for all potential confounders, the prevalence of suffering bullying (aPR 1.07, 95% CI: 1.01–1.13) and discrimination (aPR 1.52, 95% CI: 1.00–2.32) was higher among individuals with a high level of PE. Regarding the demand/control variables, positive associations were also found for low control (aPR 1.59, 95% CI: 1.30–1.96) and passive work (aPR 1.60, 95% CI: 1.23–2.08) in fully adjusted models.

Figure 2 and Figure 3 illustrate the prevalence of experiencing psychosocial hazards according to gender and level of PE (see also Table A1 in the Appendix A). Overall, the figures show that women experience more psychosocial hazards than men. Women report a higher level than men for seven out of ten hazards (not including the collapsed discrimination variable): violence or threats (11.5% vs. 10.3%), sexual harassment (15.7% vs. 3.3%), bullying (12.9% vs. 5.4%), gender discrimination (16.6% vs. 1.6%), age discrimination (12.9% vs. 6.5%), high demands (24.4% vs. 11.4%), and high strain (15.7% vs. 9.2%). The only three indicators where men report a higher level are discrimination due to ethnicity (7.6% vs. 5.1%), low control (58.7% vs. 48.8%), and passive work (49.5% vs. 33.2%).

Comparing levels of PE, the prevalence of experiencing bullying, discrimination (any), low control, and passive work is higher in high PE for both men and women. When it comes to sexual harassment, high PE is associated with an increased prevalence among men but not among women. Regarding high strain, the situation is reversed: high PE is associated with an increased prevalence among women but not men. There is no increased prevalence of high demands for men or women in high PE.

## 4. Discussion

### 4.1. Main Findings

Overall, our findings confirmed the research hypothesis that workers in high PE more frequently report experiences of bullying, discrimination, low control, and passive work, compared to workers in low PE. In addition, workers in high PE experience high strain more often than workers in low PE, although the association was not significant in the fully adjusted model. Contrary to our hypothesis, workers in high PE did not experience violence or threats and sexual harassment to a larger extent compared to workers in low PE nor was there a clear association with high demands. Finally, there were differences according to gender, where women in high PE experienced a higher level of psychosocial hazards compared to men.

### 4.2. Exposure to Violence or Threats, Sexual Harassment, Bullying, and Discrimination

The findings in this study suggested that individuals in high PE were more likely to report bullying and discrimination compared to individuals in low PE.

The results of the study did not, however, show that level of PE was associated with violence or threats, or sexual harassment. This lack of association is not in line with previous research. A recent study on PE (not using a validated measure of PE) and health-related outcomes in 28 European countries found that workers in PE were almost twice as likely to experience violence or harassment compared to non-precarious employed workers [16]. In the Swedish context, Vaez and colleagues [28] found that exposure to violence and threats was higher among temporary and part-time employees compared to permanent and full-time employees, and an Australian study found that sexual harassment was more common among temporary, casual, and self-employed workers compared to permanent full-time employees [29]. Multiple reasons could explain our differing results. First, a possible explanation for this inconsistency may be the limited sample size in our study. Further, the sample size was too small to explore occupation to a greater extent, which has been suggested as an important factor for sex discrimination and sexual harassment [20,30]. Previous research has shown that exposure to violence or threats and sexual harassment are likely to be more common in certain economic sectors, such as service, sales, and health care [28,29]. Moreover, workplace gender-dominance and norms surrounding the work role have been described to influence the prevalence of sexual harassment and sex discrimination at the workplace [20,30]. As such, future studies may benefit from having larger sample size that would allow one to take a closer look at occupations and sectors of the study participants. Another possible explanation is that both comparison groups in this study are considered to be in precarious employment, whereas other studies have compared precarious to non-precarious employees.

When it comes to bullying and discrimination, the study found a pattern of increasing prevalence in high PE. This is in line with previous research where bullying has been linked to certain PE dimensions, such as temporary employment [31]. A possible explanation could be that employees are more or less vulnerable according to their position on a core–periphery spectrum, an idea that was developed by Aronsson and colleagues [32]. According to this idea, precarious employees could become targets because of their more vulnerable position compared to non-precarious employees [32] or their ‘outsider status’ [33]. It is also possible that employers of employees in PE-relations are less aware of their legal obligations to ensure a safe work environment [34] or that precarious employees do not dare to report bullying or discrimination to the management because of their perceived vulnerability.

### 4.3. Exposure to High Demands and Low Control

This study found that the level of PE was associated with experiencing low control, an association that remained after controlling for sociodemographic background. Regarding high demands, the pattern was less clear, and the association was not significant.

When grouping the variables to measure associations with the four job types in the job strain model (high/low job strain and active/passive work) [24], we found a significant association with passive work, which is the combination of low demands and low control. This was only partly in line with the research hypothesis, where we would have expected an association with high demands and consequently with high job strain (the combination of high demands and low control). Although there is a pattern of increasing prevalence of high strain in the high PE-group, the association did not remain significant in the fully adjusted model. This may be explained by the construction of the variables, where ‘demands’ was derived from the two items “Is your work so stressful that you do not have time to talk or even think about something other than work?” and “Does work require your full attention and concentration?”. When looking at the items separately, we find that respondents in higher PE are slightly more likely to report that their work situation is stressful (35.5% vs. 32%) but less likely to report that their work requires their full attention and concentration (35.5% vs. 47.2%). There was actually a significant negative association between level of PE and high demands on attention and concentration (PR 0.76, CI 0.60–0.98) (see Table A2 in the Appendix A). This unexpected reverse pattern is in line with findings in the Swedish Work Environment Survey (AMU) of 2017, where women in temporary employment reported high demands on attention and concentration less often than women in permanent employment (for men, there was no difference depending on employment type in the AMU) [35]. This suggests that individuals in PE may be exposed to a lack of control, but rather than stress from high demands, PE may increase exposure to routine or tedious work—something that fits better with the definition of passive work (low demands and low control). In line with the interpretation that PE is more likely to increase exposure to passive work than high strain work, the AMU 2017 found passive work to be much more common among temporary employees compared to permanent employees (33% vs. 18%). There was no such difference when it came to high-strain work, where the AMU 2017 found 25% and 27% of temporary and permanent employees, respectively [35].

### 4.4. Gender Differences

As for gender differences, experiences differ for women and men in PE depending on the type of hazard. The data showed that women in our sample, compared to men, were five times as exposed to sexual harassment, more than twice as exposed to bullying, more than 10 times as exposed to gender discrimination, and almost twice as exposed to age discrimination. Women in our sample were also more exposed to high demands and high-strain work, whereas men, on the other hand, reported being discriminated due to ethnicity and more of low control and passive work (Figure 2 and Figure 3). However, when comparing low and high PE for men and women separately, the differences in prevalence were unexpectedly small for several of the outcomes. Horizontal and vertical gender segregation of the Swedish labour market, where women and men are engaged in different occupations and different positions, are likely to influence the differing patterns [17]. Previous research has found the prevalence and nature of work stress to differ depending on sector or occupation [18,36], and some studies have suggested that women and men experience demands and control differently [21,37]. We were unable to explore this further due to the limited sample size, but the gendered work context should be considered in future studies on precarious employment with larger samples.

### 4.5. Strengths and Limitations of the Study

This is one of the first studies to explore the psychosocial hazards in relation to a multidimensional construct of PE. The use of EPRES-Se to operationalize PE provides us with a more relevant and valid measure compared to what has been used in previous studies. In surveys that are otherwise used for studying work-related health, such as the European Survey on Working Conditions [38] or the Swedish Work Environment Survey (AMU) [39], there are only proxy indicators (such as contract type) available for measuring PE, which may be a limitation for the validity of the construct. Moreover, the response rate in surveys is generally low [40], especially for some sociodemographic profiles that are overrepresented in PE (e.g., foreign background and young age) [41]. The PREMIS survey had a relatively good response rate of 62%, which can be compared to the response rate in the AMU survey which was 42% in 2017 [35]. In addition, certain dimensions of PE are not easily traced in available registers (e.g., rights and benefits) which poses a challenge to register-based research on PE. Additionally, the setting for some PE relations is the grey zone between the formal and informal economy, and this activity may not be picked up in registers.

Nevertheless, there are some limitations that need to be mentioned. The sample size is relatively small which restricted the possibility to perform stratified analyses on axes of inequality such as gender and age. The EPRES instrument [42] was originally developed to measure precariousness in the general working population, and the effects observed in this study may be underestimated due to the selected sample. If the full range from no PE to high PE had been included, we may have seen larger differences and more significant prevalence ratios. Although not essential for the aim of this study, such a comparison could have been interesting.

The PREMIS survey is a unique source of data for studies on PE, but it does not investigate psychosocial hazards in depth. For example, instruments have been developed elsewhere to measure psychosocial hazards (e.g., the NAC-R for measuring bullying [43] and the JCQ [44] for measuring job demands/control), but these validated instruments are not included in the PREMIS survey.

There may be unmeasured variables that confound the results, for example, health, disability, or even personality traits. Certain health conditions or personality traits may be associated with not wanting or not obtaining secure standard employment and also associated with the likelihood to report being discriminated and stressed, etc. Self-report data may also be biased due to, e.g., recall and social desirability bias [45]. Additionally, common method bias may have increased the magnitude of the associations observed in this study [46]. Given the cross-sectional design of the study, we cannot rule out inverse associations or make any causal interpretations, but the results may be useful for generating hypotheses in future longitudinal research.

### 4.6. Generalisability

The PREMIS data are comprised of only non-standard employees in Stockholm County. The non-probability sampling technique has granted access to sociodemographic profiles that are underrepresented in national surveys (e.g., foreign background, young age and low education) [41] but at the same time limited the generalisability of the results. External validity was however not the main goal, since the scope was to study the differences according to the level of PE within this sample. However, the use of EPRES-Se makes the results comparable to existing and future studies.

Moreover, since the items from the PREMIS survey that are used to operationalize work environment hazards have very similar equivalents in the Swedish Work Environment Survey [35], it is possible to compare our data to a representative sample of Swedish workers.

In addition, the characteristics of the study sample are in line with previous studies exploring PE. High PE was more common among workers with low education, manual occupations, born outside of Sweden, and younger age [47]. An unexpected finding was that there were slightly more men than women in the high PE group. This finding may be related to the high proportion of young study participants in our sample, since previous research has found smaller gender differences in young populations [47].

## 5. Conclusions

The study finds that individuals in high PE (compared to low PE) are more likely to experience bullying and discrimination, as well as low control and passive work.

Further, the study finds that women in PE are generally more exposed than men to psychosocial hazards. Longitudinal research using larger and representative samples are needed to confirm these findings. Future research should also look into the associations between PE and specific psychosocial hazards and their connection to health and health inequalities.

## Figures and Tables

**Figure 1 ijerph-18-11218-f001:**
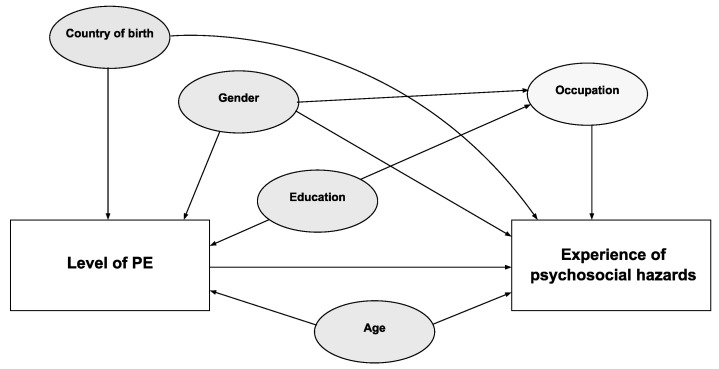
DAG (directed acyclic graph) for the effect of level of precarious employment (PE) on psychosocial hazards.

**Figure 2 ijerph-18-11218-f002:**
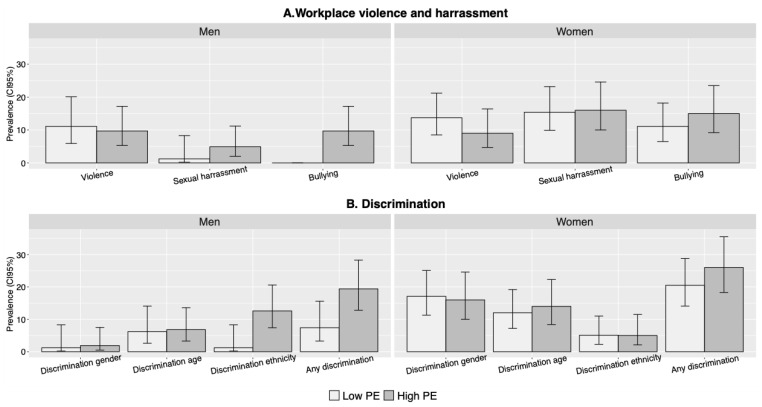
Prevalence of workplace violence, sexual harassment, and discrimination, according to gender and level of PE. (*n* = 401).

**Figure 3 ijerph-18-11218-f003:**
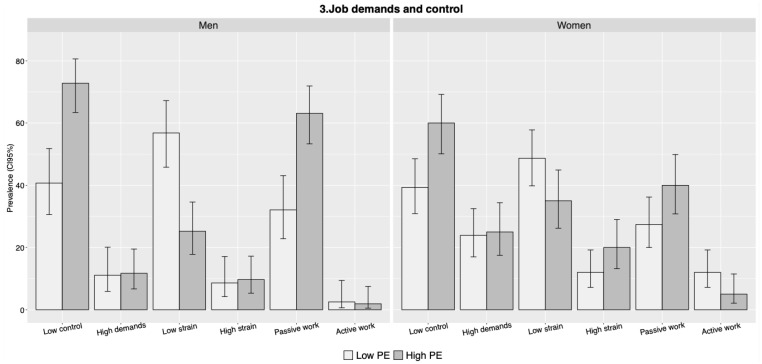
Prevalence of hazards related to demands and control, according to gender and level of PE (*n* = 401).

**Table 1 ijerph-18-11218-t001:** Characteristics of the study population by level of precarious employment (PE). PREMIS, Stockholm County, 2016–2017 (*n* = 401).

	Low PE			High PE			Total	
	*n*	(%)	CI 95%	*n*	(%)	CI 95%	*n*	(%)
Gender								
Men	81	(40.9%)	34.3–47.9	103	(50.7%)	43.9–57.6	184	(45.9%)
Women	117	(59.1%)	52.1–65.7	100	(49.3%)	42.4–56.1	217	(54.1%)
Age								
18–24	38	(19.2%)	14.3–25.3	81	(39.9%)	33.4–46.8	119	(29.7%)
25–29	94	(47.5%)	40.6–54.5	82	(40.4%)	33.8–47.3	176	(43.9%)
30–35	41	(20.7%)	15.6–26.9	20	(9.9%)	6.4–14.8	61	(15.2%)
36–62	25	(12.6%)	8.7–18.0	20	(9.9%)	6.4–14.8	45	(11.2%)
Education								
≤High school	68	(34.9%)	28.5–41.8	88	(45.6%)	38.7–52.7	156	(40.2%)
Higher education, ≤2 y	44	(22.6%)	17.2–29.0	51	(26.4%)	20.7–33.1	95	(24.5%)
Higher education, ≥3 y	83	(42.6%)	35.8–49.6	54	(28.0%)	22.1–34.7	137	(35.3%)
Country of birth								
Sweden	167	(84.8%)	79.0–89.2	155	(77.1%)	70.8–82.4	322	(80.9%)
Non-Sweden	30	(15.2%)	10.8–21.0	46	(22.9%)	17.6–29.2	76	(19.1%)
General health								
Good or very good	146	(73.7%)	67.1–79.4	127	(62.6%)	55.7–69.0	273	(68.1%)
Less than good	52	(26.3%)	20.6–32.9	76	(37.4%)	31.0–44.3	128	(31.9%)
Occupational social class								
Manual	92	(48.9%)	41.8–56.1	118	(59.3%)	52.3–65.9	210	(54.3%)
Non-manual	96	(51.1%)	43.9–58.2	81	(40.7%)	34.1–47.7	177	(45.7%)

This table contains missing values: Education: 13 missing, Country of Birth: 3 missing, and Occupation: 14 missing.

**Table 2 ijerph-18-11218-t002:** Workplace violence, sexual harassment, bullying, discrimination, and job demands/control by low (reference group) and high precarious employment (PE). (*n* = 401).

Psychosocial Hazards	Low PE (*n* = 198)			High PE (*n* = 203)				
	*n* ^b^	(%)	PR = Ref.	*n* ^b^	(%)	PR_1_ (CI 95%)	PR_2_ (CI 95%)	PR_3_ (CI 95%)
Violence or threats	25	(12.6%)	-	19	(9.4%)	0.97 (0.92–1.03)	0.97 (0.92–1.03)	0.96 (0.91–1.02)
Sexual Harassment	19	(9.6%)	-	21	(10.3%)	1.01 (0.95–1.06)	1.01 (0.96–1.07)	1.01 (0.96–1.07)
Bullying	13	(6.6%)	-	25	(12.3%)	1.05 (1.00–1.11)	1.07 (1.01–1.12)	1.07 (1.01–1.13)
Discrimination ^a^	30	(15.2%)	-	46	(22.7%)	1.50 (0.99–2.27)	1.57 (1.03–2.40)	1.52 (1.00–2.32)
High demands	37	(18.7%)	-	37	(18.2%)	0.98 (0.65–1.47)	1.17 (0.77–1.78)	1.10 (0.71–1.70)
Low control	79	(39.9%)	-	135	(66.5%)	1.67 (1.37–2.03)	1.68 (1.37–2.05)	1.59 (1.30–1.96)
High strain	21	(10.6%)	-	30	(14.8%)	1.39 (0.83–2.35)	1.72 (1.01–2.92)	1.54 (0.89–2.66)
Low strain	103	(52.0%)	-	61	(30.0%)	0.58 (0.45–0.74)	0.56 (0.44–0.72)	0.60 (0.46–0.78)
Active work	16	(8.1%)	-	7	(3.4%)	0.43 (0.18–1.02)	0.49 (0.20–1.19)	0.52 (0.21–1.33)
Passive work	58	(29.3%)	-	105	(51.7%)	1.77 (1.37–2.28)	1.65 (1.28–2.13)	1.60 (1.23–2.08)

^a^ Discrimination is the sum of discrimination due to age, gender, and ethnicity. ^b^ prevalence (and percentage) of experiencing psychosocial hazards. PR = prevalence ratio. PR_1_ = crude, PR_2_ = adjusted for gender and age, and PR_3_ = adjusted for gender, age, education, and country of birth.

## Data Availability

De-identified participant data from the PREMIS study cannot be made publicly available for ethical and legal reasons. It could however be made available to researchers who meet the criteria for access to confidential data, after approval from the Regional Ethics Committee of Stockholm. If interested, contact the corresponding author: signild.kvart@ki.se.

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
