# Peer review of "Precarious Employment and Psychosocial Hazards: A Cross-Sectional Study in Stockholm County"

_ijerph, 2021, doi:10.3390/ijerph182111218_

Round 1
Reviewer 1 Report
Thank you for the opportunity to read this manuscript, which I believe has significant importance for the labor market development and different stakeholders in the field, in Sweden as well as other countries. Before presenting suggestions that would improve the manuscript, I want to point out that this is a well written manuscript with interesting findings.
Abstract
The abstract is well written, but lacks information that the study is based on survey data and register data.
- Introduction
The introduction gives a short overview of the necessary aspects to report. However, line 55-65 is hard to follow, and need to be clarified to improve the readability. For example, would it be possible to use PE, instead of returning to the concept of non-standard employment? Is job strain an example of psychosocial work environment? Is threats, bullying, harassment and discrimination examples of occupational violence, or something else?
Your focus is the association between PE and psychosocial work environment hazards, but I think you should mention previous research about the link between psychosocial work environment hazards and adverse health, preferably with a gender perspective. Health inequality and social determinants of health is after all concepts that you address in the abstract/key words.
Please, give examples of what lack of rights and protection means (line 55). Sickness benefits, vacation?
- Materials and Methods
Regarding the data collection, you refer to previous publications for further information. However, some information about how you identified the study participants would be appropriate, that to say, how did you know who to send your survey to?
Regarding statistical analysis, why did you choose Poisson family? Not all readers (including me) are familiar with this method of analysis, so it would be great with a short description of what it contributes with.
- Results
The results are clearly written and easy to follow. However, some main text is placed within the Table 1 heading (line 195-198), please revise this.
- Discussion
The discussion is well written and gives a deeper understanding of your findings. First, a few minor comments:
- I think you should start with discussing the findings, before presenting the strengths and limitation. I would suggest to move Strengths and limitation after the heading Gender differences, and before Generalisability.
- You discuss (line 298-306 + line 361-368) that it was not possible to include occupation in terms of e.g. male or female dominated sectors. This is important, and also an aspect to add as a limitation after gender and age? (line 264).
- Some references are not following the journals reference system, please revise this (e.g. line 111, 292, 314), and make sure all references are included in the reference list.
After reading the manuscript and the discussion, there are two questions and “red lines” that remain unanswered.
1) You discuss that it would have been interesting to include health in your analysis, at the same time as you do have such data from the survey. This raise the question why you did not include health as a variable in the analysis? Please provide an argument for this, rather than avoiding the health aspect in the manuscript. Health inequality and social determinants of health is aspects that you mention in the abstract/key words (and should be mentioned!).
2) The results show that women to a lower extent than men experience discrimination due to ethnicity. This is something to further elaborate. Previous research has showed that people who experience discrimination based on several grounds may have difficulties to distinguish the specific cause behind discrimination, which was a strong reason behind the development of intersectional perspectives in research. Could one reason be that women with foreign background in your study have harder to distinguish that they are subjected to racism, compared to men with foreign background? Or is the percentage of men with foreign background larger than the percentage with women with foreign background?
Author Response
Please see the attachment, thank you.

Reviewer 2 Report
This is an interesting study that evaluated the association between precarious employment and occupational risks, particularly psychosocial hazards. The paper contributes to the knowledge on the theme, as PE has become a global concern during the past decades due to the reestructuring of production and neoliberal policies. I believe very few aspects should be addressed/revised before publication, as listed below:
Results section: Please review the third line in table 2. The result looks strange, as bullying was almost twice more frequent among high PE workers, but the PR showed was 1.05 (Bullying 13 (6.6%) - 25 (12.3%) 1.05 (1.00–1.11)). Is that correct?
Discussion: Although the association between high PE and high strain was not statistically significant in the fully adjusted model, the prevalence of high strain was notably higher among those with high PE. The lack of statistical significance might be due to lack of power... On the other hand, the prevalence of low strain was 40% lower among those with high PE. This interesting result was not discussed in depth, in it is something that could be highlighted by the authors, if appropriate.
Author Response
Please see the attachment, thank you.
